# Sublobar Resection Versus Lobectomy for Small (≤3 cm) NSCLC with Visceral Pleural Invasion: A Propensity-Score-Matched Survival Analysis from a Nationwide Cohort

**DOI:** 10.3390/cancers17121990

**Published:** 2025-06-14

**Authors:** Xu-Heng Chiang, Chi-Jen Chen, Chih-Fu Wei, Yu-An Zheng, Ching-Chun Lin, Mong-Wei Lin, Chun-Ju Chiang, Wen-Chung Lee, Jin-Shing Chen, Pau-Chung Chen

**Affiliations:** 1Department of Medical Education, National Taiwan University Hospital, Taipei 100225, Taiwan; ntuhxhc@ntu.edu.tw; 2Department of Surgery, National Taiwan University Hospital and National Taiwan University College of Medicine, Taipei 100233, Taiwan; a679nn@ntuh.gov.tw (Y.-A.Z.); mwlin@ntu.edu.tw (M.-W.L.); 3Institute of Epidemiology and Preventive Medicine, College of Public Health, National Taiwan University, Taipei 106319, Taiwan; d08849021@ntu.edu.tw (C.-J.C.); ruru.chiang@cph.ntu.edu.tw (C.-J.C.); wenchung@ntu.edu.tw (W.-C.L.); 4Department of Environmental and Occupational Medicine, National Taiwan University Hospital Yun-Lin Branch, Yun-Lin 640203, Taiwan; y09486@ms1.ylh.gov.tw; 5Institute of Environmental and Occupational Health Sciences, College of Public Health, National Taiwan University Hospital, Taipei 100025, Taiwan; chingchun@ntu.edu.tw; 6Taiwan Cancer Registry, Taipei 10055, Taiwan; 7Institute of Health Data Analytics and Statistics, College of Public Health, National Taiwan University, Taipei 106319, Taiwan; 8Department of Public Health, National Taiwan University College of Public Health, Taipei 10055, Taiwan; 9Department of Environmental and Occupational Medicine, National Taiwan University Hospital and National Taiwan University College of Medicine, Taipei 100233, Taiwan; 10National Institute of Environmental Health Sciences, National Health Research Institutes, Miaoli 35053, Taiwan

**Keywords:** non-small-cell lung cancer (NSCLC), visceral pleural invasion (VPI), sublobar resection, lobectomy, propensity score matching, early lung cancer

## Abstract

For clinicians treating small-sized non-small-cell lung cancers with visceral pleural invasion, choosing the best surgical approach—either less-extensive sublobar resection or standard lobectomy—has been challenging due to mixed evidence. This study investigated this issue using a large patient database and found that lung-cancer-specific survival rates were comparable between both surgical methods for these specific tumors (≤3 cm without lymph node spread). This suggests sublobar resection can be an equally effective, potentially less invasive, treatment option for appropriately selected patients, offering valuable guidance for surgical decision-making, though further confirmation from future studies is encouraged.

## 1. Introduction

Lobectomy has long been considered the standard surgical approach for curative-intent resection of early-stage non-small-cell lung cancer (NSCLC) [1]. However, this paradigm has been increasingly challenged over the past decade. Several landmark randomized controlled trials, notably the JCOG0802/WJOG4607L study and the CALGB 140503 trial, have provided robust evidence that sublobar resection (SLR), encompassing segmentectomy and wedge resection, can achieve non-inferior or equivalent oncological outcomes compared to lobectomy for select patients with small (typically defined as ≤2 cm), peripheral NSCLCs [2,3]. These findings have ushered in an era where SLR is a valid alternative for appropriately selected early-stage lung cancers.

Nevertheless, surgical decision-making in NSCLC is complex and extends beyond tumor size alone. A multitude of clinicopathological factors significantly influence patient prognosis and guide the selection of the optimal surgical strategy [4]. For instance, the radiological appearance of a tumor, such as the proportions of the ground-glass opacity and solid components, is often considered; tumors with a predominant ground-glass opacity component might be deemed suitable for SLR, whereas highly solid tumors, even if small, may prompt consideration for lobectomy due to potentially more aggressive biology [5,6]. Among these prognostic factors, visceral pleural invasion (VPI) is well-established as an indicator of adverse outcomes in NSCLC [7]. The presence of VPI, defined as tumor extension beyond the elastic layer of the visceral pleura, is associated with a higher risk of recurrence and poorer survival, often leading to an upstaging of the pathological tumor classification, e.g., a T2a designation even for tumors ≤ 3 cm according to the 8th edition AJCC staging manual [8,9].

This creates a critical clinical conflict. Whereas recent trials endorse SLR for selected small NSCLCs [2,3], the inherent biological aggressiveness signified by VPI raises the question of whether lobectomy is necessary for adequate oncological control. The central issue is whether the negative impact of VPI overrides the potential sufficiency of SLR in the context of small tumor size. Retrospective studies comparing lobectomy and SLR specifically for small (e.g., ≤2 cm or ≤3 cm), VPI-positive NSCLCs have yielded conflicting results. Some large database analyses suggest superior survival outcomes with lobectomy [10], whereas others, including meta-analyses or different database studies, indicate that SLR—particularly segmentectomy or when used for tumors ≤ 2 cm—may achieve comparable survival rates [11,12]. Consequently, owing to these inconsistent findings, often derived from retrospective data with inherent limitations, a definitive consensus on the optimal surgical extent is lacking. This represents a significant knowledge gap regarding the appropriate management strategy for patients presenting with small-sized, VPI-positive NSCLCs [10,13].

To address this knowledge gap, the present study leveraged the comprehensive data resources of the Taiwan Cancer Registry (TCR). The TCR is a rigorously maintained, nationwide, population-based database known for its high quality and completeness, providing a valuable platform for real-world evidence generation [14,15]. By analyzing survival outcomes in a large representative cohort of patients from the TCR diagnosed with small-sized NSCLC and confirmed VPI, we aimed to compare the effectiveness of lobectomy and SLR. This study seeks to provide crucial evidence to inform surgical decision-making for this challenging patient population.

## 2. Materials and Methods

### 2.1. Study Population

Data for this retrospective cohort study were retrieved from the TCR, a validated nationwide population-based database [14,15]. We initially identified 36,560 patients diagnosed with NSCLC from 1 January 2011 to 31 December 2018. Patients were selected according to a sequential algorithm, which is depicted in Figure 1. The exclusion criteria included a lack of surgical intervention, pathologically confirmed tumor size greater than 3 cm, absence of visceral pleural invasion (PL0) or presence of extensive pleural involvement classified as PL3 (T3 disease), pathologically confirmed nodal or distant metastasis (N+ or M1), involved surgical margins, and duplicate or incomplete records unsuitable for analysis. The final analytic cohort comprised 2460 patients with pathologically staged T2aN0M0 NSCLC, characterized by tumors ≤ 3 cm in size with VPI (corresponding to PL1 or PL2). These patients were then categorized according to the surgical procedure performed: SLR (n = 624) or lobectomy (n = 1836).

### 2.2. Propensity Score Matching

To minimize potential selection bias and confounding effects due to differences in baseline characteristics between the SLR and lobectomy groups, we performed propensity score matching (PSM). Propensity scores, representing the estimated probability of receiving SLR or lobectomy given the observed covariates, were calculated using a multivariable logistic regression model. Based on clinical relevance and previous literature [1,5,8], the covariates included in the model were as follows: patient age, sex, self-reported smoking history, tumor size, pathological T stage (pT), histological type, tumor differentiation grade, lymphovascular invasion (LVI), and VPI. Patients who underwent SLR were matched 1:1 to those who underwent lobectomy using a greedy nearest-neighbor matching algorithm without replacement. A caliper width of 0.01 standard deviations of the logit of the propensity score was applied during matching. Exact matching was not enforced for any covariate. The goodness-of-fit and discriminative ability of the logistic regression model were evaluated using the Hosmer–Lemeshow test and the c-statistic. Furthermore, the distribution of propensity scores before and after matching was visually inspected to assess the balance achieved (Appendix A).

### 2.3. Surgical Modalities and Main Outcomes

The primary exposure variable in this study was the type of curative-intent pulmonary resection performed. Patients included in the final cohort, all having undergone complete (margin-negative) resection, were categorized into two groups based on the surgical procedure documented in the TCR: lobectomy or SLR. The primary endpoint for evaluating treatment effectiveness was lung-cancer-specific survival (LCSS). The use of LCSS as the primary endpoint allowed for a focused assessment of surgical efficacy against cancer-related death, mitigating confounding from mortality due to other causes (competing risks), which is particularly relevant in population-based studies [16]. LCSS was defined as the time from definitive surgical resection to death attributed to lung cancer, censoring for death from other causes [17]. Survival status, date of death, and cause of death information were ascertained by linking the TCR records with data obtained from the official Taiwanese national death registry database.

### 2.4. Statistics

All statistical analyses were performed using R version 4.5.0 (R Foundation for Statistical Computing, Vienna, Austria) and SAS 9.4 (SAS Institute, Cary, NC, USA). A two-sided *p*-value < 0.05 was considered statistically significant.

Descriptive statistics were generated to summarize patient demographics and baseline clinicopathological characteristics for the study population prior to matching and the propensity-score-matched cohort. Continuous variables are presented as mean ± standard deviation (SD). Categorical variables are presented as frequencies (n) and percentages (%). Baseline characteristics between the lobectomy and SLR groups were compared using Student’s *t*-tests or Mann–Whitney U tests for continuous variables, and chi-squared tests for categorical variables.

LCSS probabilities were estimated using the Kaplan–Meier method, and survival curves were generated for the lobectomy and SLR groups, before and after matching. Differences between the Kaplan–Meier survival curves were compared using the log-rank test.

To evaluate the independent association between the type of surgical procedure (SLR versus lobectomy) and LCSS, Cox proportional hazards regression models were utilized. Hazard ratios (HRs) and their corresponding 95% confidence intervals (CIs) were estimated. Both unadjusted (crude) and multivariable-adjusted Cox models were developed.

## 3. Results

### 3.1. Baseline Characteristics

A total of 2460 patients with surgically resected pT2a (≤3 cm with VPI) N0M0 NSCLC met the inclusion criteria and constituted the initial cohort for analysis before PSM (Table 1). Of these, 624 (25.4%) underwent SLR and 1836 (74.6%) underwent lobectomy. The mean age of this overall cohort was 63.7 ± 10.4 years. Most patients were female (1415 patients, 57.5%). Most patients were never smokers (1791 patients, 72.8%). Histologically, adenocarcinoma was overwhelmingly the most common subtype, identified in approximately 90% of cases (2204 patients, 89.6%). Regarding the extent of VPI, 1873 patients (76.1%) were classified as PL1, and 587 (23.9%) as PL2. During the follow-up period documented in the registry for this cohort (2011–2024), disease recurrence was observed in 439 patients, comprising 73 (3.0%) cases of locoregional recurrence and 366 (14.9%) cases of distant recurrence.

### 3.2. PSM for LCSS Analysis

PSM successfully yielded 523 matched pairs of patients who underwent either SLR or lobectomy. As shown in Table 1, baseline demographics and clinicopathological characteristics were well-balanced between the SLR and lobectomy groups after matching. No statistically significant differences were observed for key variables including age, sex, body mass index, smoking status, tumor size categories, histological type, tumor differentiation grade, LVI, or VPI status (all *p* > 0.05, Table 1). Notably, tumor laterality was not included as a covariate in the PSM model due to its perceived lesser impact on prognosis and, consequently, remained significantly different between the groups after matching (*p* < 0.001, Table 1). The resulting matched cohort (n = 1046) had a mean age of 66.9 ± 9.9 years, and comprised 573 (54.8%) female patients (Table 1).

### 3.3. LCSS Before and After Matching

LCSS was first compared between the SLR and lobectomy groups in the initial cohort prior to matching (Figure 2). This analysis revealed that patients who underwent lobectomy had significantly better LCSS than those who received SLR (Figure 2A, *p* = 0.01). When stratified by the extent of visceral pleural invasion, a similar significant survival advantage for lobectomy was observed among patients with PL1 invasion (Figure 2B, *p* = 0.009). In contrast, within the subgroup of patients with PL2 invasion, no statistically significant difference in LCSS was detected between the two surgical approaches (Figure 2C, *p* = 0.14).

Following PSM, which yielded 523 pairs of patients with balanced baseline characteristics, Kaplan–Meier survival analysis was repeated (Figure 3). In the overall matched cohort, the survival curves for SLR and lobectomy were closely aligned, and the previously observed significant difference was no longer present (Figure 3A, *p* = 0.21). Stratified analyses within the matched cohort also showed no significant differences in LCSS between SLR and lobectomy, both for patients with PL1 invasion (Figure 3B, *p* = 0.11) and for those with PL2 invasion (Figure 3C, *p* = 0.94). Furthermore, an analysis comparing segmentectomy and wedge resection within this matched cohort revealed no significant difference in lung-cancer-specific survival between these two sublobar approaches (Appendix A).

### 3.4. Cox Proportional Hazards Models for LCSS

The results of the Cox proportional hazards regression analyses for LCSS are presented in Table 2. In the univariate analysis, lobectomy was associated with significantly better LCSS, compared to that for SLR (HR 0.66, 95% CI 0.47−0.91, *p* = 0.011). However, after adjusting for other covariates in the multivariate Cox model, the type of surgical procedure (lobectomy vs. SLR) was no longer significantly associated with LCSS (adjusted HR 0.75, 95% CI 0.52−1.08, *p* = 0.124). The multivariate analysis identified several independent prognostic factors significantly associated with worse LCSS: age greater than 75 years (adjusted HR 1.97, 95% CI 1.40−2.77, *p* < 0.001), presence of PL2 VPI (PL2 vs. PL1; adjusted HR 2.04, 95% CI 1.50−2.77, *p* < 0.001), and presence of LVI (adjusted HR 5.21, 95% CI 2.84−9.57, *p* < 0.001).

## 4. Discussion

The management of early-stage NSCLC has evolved significantly, with recent landmark trials demonstrating the oncological non-inferiority of SLR compared to lobectomy for selected patients with small (≤2 cm), peripheral tumors [2,3]. However, the optimal surgical extent for small tumors exhibiting adverse pathological features, such as VPI, remains a subject of debate [13,18]. This study, utilizing a large, nationwide, population-based cohort in Taiwan, investigated this specific scenario. Our primary finding, derived from propensity-score-matched analysis and multivariate Cox regression, is that after adjusting for significant prognostic factors, SLR and lobectomy provide comparable LCSS for patients with pT2a (≤3 cm with VPI) N0M0 NSCLC. While lobectomy showed a survival advantage in the unmatched cohort and univariate analysis, this difference was attenuated and lost statistical significance after balancing baseline characteristics and accounting for confounding factors. This suggests that VPI, in the absence of nodal or distant metastasis for tumors ≤ 3 cm, may not independently necessitate more extensive resection, such as lobectomy, to achieve favorable long-term cancer-specific outcomes.

Our findings align with those of several recent studies suggesting that SLR, particularly segmentectomy, can be a viable option, even in the presence of VPI, for appropriately selected small NSCLCs. For instance, Eguchi et al. [12], using the US National Cancer Database (NCDB) and focusing on clinical T1a-bN0 tumors found to have VPI postoperatively, reported no significant difference in overall survival between segmentectomy and lobectomy after PSM. Similarly, a meta-analysis by Dai et al. [11] indicated that while SLR generally showed worse outcomes for VPI-positive tumors ≤ 3 cm, the difference was not statistically significant for tumors ≤ 2 cm, and segmentectomy achieved comparable survival to lobectomy across the ≤3 cm VPI-positive group. Other studies focusing on segmentectomy versus lobectomy for VPI-positive small NSCLCs have also reported comparable long-term results, further supporting the potential adequacy of less-extensive resection in this subgroup [19,20]. These concordant findings, despite originating from different databases (TCR, NCDB, SEER) and multicenter studies, and employing varied methodologies, strengthen the hypothesis that anatomical SLR might suffice for small VPI-positive tumors if complete resection and adequate nodal assessment are achieved.

Conversely, our results contrast with those of other investigations that concluded that lobectomy offers superior survival for small, VPI-positive NSCLCs. Zhao et al. [10], analyzing the SEER database for tumors ≤ 2 cm with VPI, found that lobectomy was associated with better long-term disease-specific and overall survival compared to those of sublobectomy, even after PSM. Another SEER-based study focusing on stage IB (≤3 cm with VPI) reported lobectomy to be superior to wedge resection, but comparable to segmentectomy [21]. Differences between these studies and ours may stem from several factors, including variations in the definition of SLR (some combined wedge and segmentectomy, whereas others analyzed them separately), differences in patient populations across databases (SEER vs. TCR), statistical methodologies (adjustment variables in Cox models, PSM implementation details), and potentially the specific endpoint analyzed (overall survival vs. LCSS) [10,22,23,24]. Furthermore, residual confounding factors, despite PSM, cannot be entirely excluded from observational studies and may contribute to disparate findings [25,26].

The finding that surgical extent (SLR vs. lobectomy) did not independently impact LCSS in our adjusted analysis, despite the established negative prognostic nature of VPI [7], warrants exploration. One plausible explanation relates to the tumor size restriction (≤3 cm) and the nature of resection. For such small lesions, even anatomical SLR (primarily segmentectomy in many centers for oncological indications) often allows for the achievement of adequate surgical margins, potentially negating the theoretical benefit of the wider parenchymal removal offered by lobectomy in terms of local control for truly localized disease [27]. Complete resection with negative margins is paramount, and perhaps more critical than the volume of tissue removed when margins are clear. Second, our stringent exclusion criteria, removing patients with any nodal involvement (N+), distant metastasis (M1), or extensive pleural disease (PL3, potentially implying parietal pleural invasion or M1a pleural seeding), ensured that the analyzed cohort represented truly localized disease despite the VPI. In such cases where the tumor cells, although invading the visceral pleura, have not demonstrably spread beyond the lung parenchyma or immediate pleura, the systemic risk might be primarily driven by factors like LVI (which remained highly significant in our multivariate analysis) rather than the local extent treatable by the resection type difference between SLR and lobectomy. The biological behavior dictated by factors like LVI, and the inherent tumor aggressiveness reflected partly by VPI status (PL2 was significantly associated with poorer LCSS than PL1) and differentiation grade, may ultimately be stronger determinants of LCSS than the choice between SLR and lobectomy in this specific, localized N0M0 context [9,28].

It is pertinent to address the decision not to include the number of lymph nodes retrieved as a covariate in our propensity-score-matching model. This choice stems from the inherent difference in lymph node yield between lobectomy and sublobar resection (SLR), which is a widely recognized characteristic of these procedures [29]. While the lobectomy group had a significantly higher mean number of lymph nodes retrieved (19.29 vs. 12.99, *p* < 0.001), the mean for the SLR group (12.99) still considerably surpasses the minimum six lymph nodes recommended by international guidelines for adequate pathological evaluation and accurate staging [30]. For our specific pT2a (≤3 cm) N0M0 cohort, where patients are clinically node-negative, the probability of this difference materially altering final staging accuracy is deemed low, a view supported by trials on similar early-stage populations [31]. Our matching strategy, therefore, focused on balancing other patient and tumor characteristics that are less directly determined by the surgical extent itself, allowing us to robustly analyze lung-cancer-specific survival as our primary outcome.

This study has several strengths. First, the use of the TCR [14,15], a high-quality, nationwide, population-based database with rigorous validation processes, provided a large and representative cohort, enhancing the generalizability of our findings to the Taiwanese population with this specific stage of NSCLC. The large sample size (initially 2460 patients in this subgroup) allowed for robust statistical analysis. Second, the use of PSM [32] effectively balanced a wide range of measured baseline covariates between the SLR and lobectomy groups, significantly reducing the potential selection bias inherent in observational studies comparing surgical procedures. Third, our focus on LCSS as the primary endpoint, using data linked to the national death registry, minimized bias from competing risks of mortality, providing a clearer assessment of the impact of surgical modality on cancer-related death [16]. Finally, by focusing specifically on the controversial subgroup of small (≤3 cm) NSCLC with VPI but without nodal or distant metastasis, this study addressed a pertinent clinical question where high-level evidence is still evolving.

Despite these strengths, our study has limitations inherent to its retrospective, registry-based design. Although the TCR provides comprehensive cancer data [14], it lacks detailed preoperative imaging information, such as the consolidation-to-tumor ratio or precise tumor location, which can influence surgical decisions and prognosis [5]. Furthermore, important perioperative details, including specific surgical techniques within SLR (wedge vs. segmentectomy breakdown), operation time, blood loss, postoperative complications, and length of hospital stay are not captured in the database. This information lack hindered the assessment of potential differences in perioperative outcomes between SLR and lobectomy. Moreover, the database does not include records of preoperative assessments, such as pulmonary function tests or detailed comorbidity indices (beyond age), which are crucial for evaluating surgical candidacy and may represent unmeasured confounders influencing the choice of surgical extent, even after PSM. These limitations, stemming from the nature of secondary data analysis, highlight the need for prospective studies or registries collecting more granular clinical, imaging, and perioperative data to fully compare the comprehensive outcomes of SLR versus lobectomy for this challenging subgroup of early-stage NSCLC.

## 5. Conclusions

In this large, population-based cohort study of patients with pathologic T2a (≤3 cm with VPI) N0M0 NSCLC, SLR demonstrated comparable LCSS to lobectomy after PSM and adjustment for prognostic factors. While surgical extent itself was not an independent predictor of LCSS in the adjusted analysis, older age, PL2 VPI, and LVI were significantly associated with worse outcomes. These findings suggest that SLR may be an important oncological alternative to lobectomy for selected patients with small (≤3 cm), node-negative NSCLC, even in the presence of VPI. However, further validation through large-scale prospective studies is warranted to confirm these results and solidify their implications for clinical practice.

## Figures and Tables

**Figure 1 cancers-17-01990-f001:**
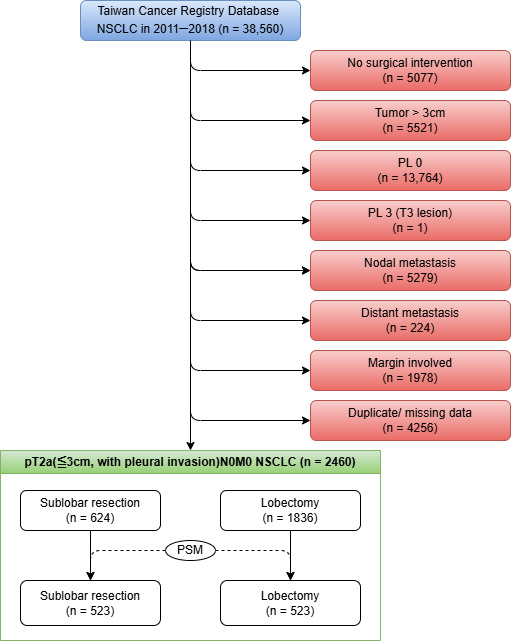
Selection algorithm.

**Figure 2 cancers-17-01990-f002:**
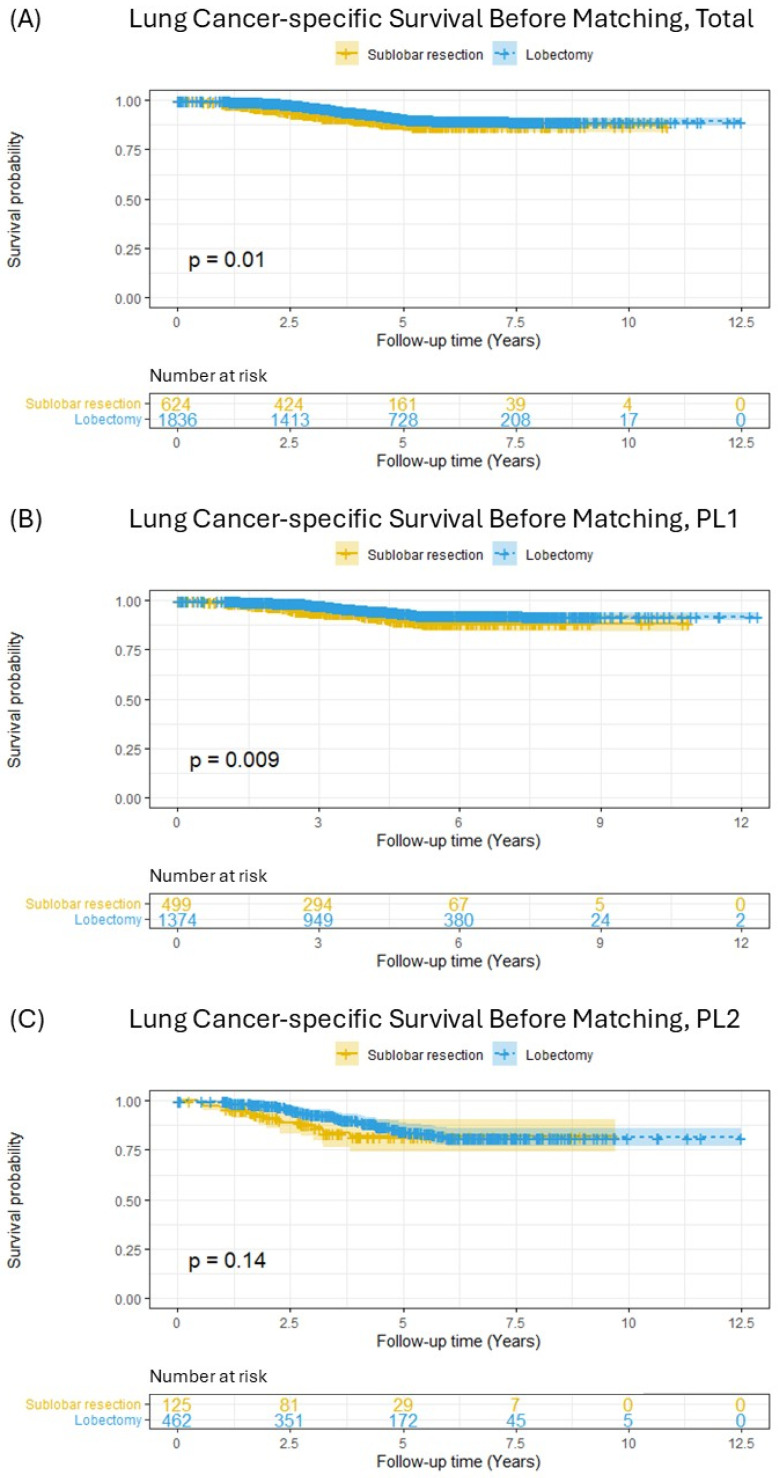
Lung-cancer-specific survival in patients with pT2aN0M0 (tumor ≤ 3 cm) non-small-cell lung cancer before matching; (**A**) total included patients, (**B**) PL1 patients, (**C**) PL2 patients.

**Figure 3 cancers-17-01990-f003:**
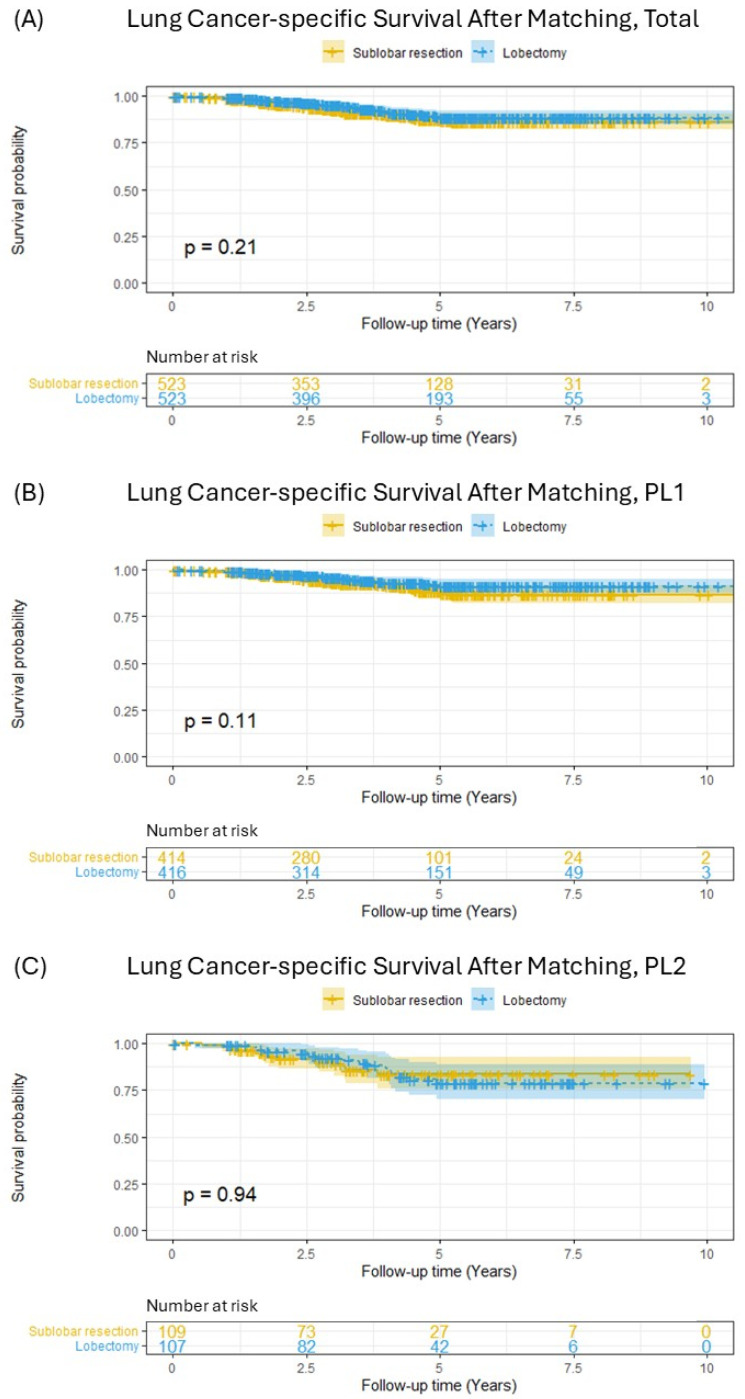
Lung-cancer-specific survival in patients with pT2aN0M0 (tumor ≤ 3 cm) non-small-cell lung cancer after matching; (**A**) total included patients, (**B**) PL1 patients, (**C**) PL2 patients.

**Table 1 cancers-17-01990-t001:** Demographic and clinical features of patients with pT2a (≤3 cm, with pleural invasion) N0M0 NSCLC that underwent sublobar resection or lobectomy.

	Before Matching				After Matching			
	Total	Sublobar Resection	Lobectomy	*p* Value	Total	Sublobar Resection	Lobectomy	*p* Value
	(n = 2460)	(n = 624)	(n = 1836)		(n = 1046)	(n = 523)	(n = 523)	
Age, yrs	63.7 (10.4)	68.0 (10.6)	62.3 (9.9)	<0.001	66.86 (9.85)	66.94 (9.97)	66.77 (9.75)	0.775
Female	1415 (57.5%)	331 (53.0%)	1084 (59.0%)	0.010	573 (54.8%)	283 (54.1%)	290 (55.4%)	0.709
BMI, kg/m^2^	24.3 (3.6)	24.6 (3.7)	24.2 (3.5)	0.034	24.41 (3.62)	24.59 (3.79)	24.23 (3.42)	0.113
Smoking status				0.003				0.893
Never smoker	1791 (72.8%)	425 (68.1%)	1366 (74.4%)		727 (69.5%)	362 (69.2%)	365 (69.8%)	
Ever smoker	669 (27.2%)	199 (31.9%)	470 (25.6%)		319 (30.5%)	161 (30.8%)	158 (30.2%)	
Laterality				<0.001				<0.001
Right	1525 (62.0%)	338 (54.2%)	1187 (64.7%)		621 (59.4%)	277 (53.0%)	344 (65.8%)	
Left	935 (38.0%)	286 (45.8%)	649 (35.3%)		425 (40.6%)	246 (47.0%)	179 (34.2%)	
Tumor size, cm				<0.001				0.683
0–1	124 (5.0%)	65 (10.4%)	59 (3.2%)		76 (7.3%)	37 (7.1%)	39 (7.5%)	
1–2	924 (37.6%)	305 (48.9%)	619 (33.7%)		516 (49.3%)	252 (48.2%)	264 (50.5%)	
2–3	1412 (57.4%)	254 (40.7%)	1158 (63.1%)		454 (43.4%)	234 (44.7%)	220 (42.1%)	
Differentiation *				0.217				0.933
Well	240 (9.8%)	71 (11.4%)	169 (9.2%)		115 (11.0%)	57 (10.9%)	58 (11.1%)	
Moderate	1572 (63.9%)	397 (63.6%)	1175 (64.0%)		664 (63.5%)	336 (64.2%)	328 (62.7%)	
Poor	585 (23.8%)	145 (23.2%)	440 (24.0%)		254 (24.3%)	123 (23.5%)	131 (25.0%)	
N/A	63 (2.6%)	11 (1.8%)	52 (2.8%)		13 (1.2%)	7 (1.3%)	6 (1.1%)	
Histology				0.001				0.615
Adenocarcinoma	2204 (89.6%)	534 (85.6%)	1670 (91.0%)		906 (86.6%)	449 (85.9%)	457 (87.4%)	
SCC	107 (4.3%)	40 (6.4%)	67 (3.6%)		55 (5.3%)	31 (5.9%)	24 (4.6%)	
Others	149 (6.1%)	50 (8.0%)	99 (5.4%)		85 (8.1%)	43 (8.2%)	42 (8.0%)	
Lymphovascular invasion	94 (3.8%)	25 (4.0%)	69 (3.8%)	0.874	39 (3.7%)	19 (3.6%)	20 (3.8%)	1.000
Visceral pleural invasion				0.011				0.939
PL1	1873 (76.1%)	499 (80.0%)	1374 (74.8%)		830 (79.3%)	414 (79.2%)	416 (79.5%)	
PL2	587 (23.9%)	125 (20.0%)	462 (25.2)		216 (20.7%)	109 (20.8%)	107 (20.5%)	
Lymph nodes examined number	17.52 (11.30)	12.70 (10.20)	19.16 (11.19)	<0.001	16.14 (11.24)	12.99 (10.39)	19.29 (11.18)	<0.001
Recurrence *				0.061				0.249
In situ	0				0			
Locoregional	73 (3.0%)	23 (3.7%)	50 (2.7%)		38 (3.6%)	20 (3.8%)	18 (3.4%)	
Distant	366 (14.9%)	80 (12.8%)	286 (15.6%)		141 (13.5%)	74 (14.1%)	67 (12.8%)	
N/A	105 (4.3%)	35 (5.6%)	70 (3.8%)		46 (4.4%)	29 (5.5%)	17 (3.3%)	
Adjuvant therapy				<0.001				0.408
CCRT	4 (0.2%)	2 (0.3%)	2 (0.1%)		2 (0.2%)	2 (0.4%)	0 (0.0%)	
Chemotherapy	831 (33.8%)	164 (26.3%)	667 (36.3%)		310 (29.6%)	146 (27.9%)	164 (31.4%)	
Radiotherapy	5 (0.2%)	2 (0.3%)	3 (0.2%)		2 (0.2%)	1 (0.2%)	1 (0.2%)	
TKI	11 (0.4%)	4 (0.6%)	7 (0.4%)		6 (0.6%)	4 (0.8%)	2 (0.4%)	

Data are presented as mean ± SD (range) or number (%). * Missing data for differentiation and recurrence are indicated by N/A. BMI, body mass index; CCRT, concurrent chemoradiotherapy; NSCLC, non-small-cell lung cancer; SCC, squamous cell carcinoma; TKI, tyrosine kinase inhibitor.

**Table 2 cancers-17-01990-t002:** Cox regression analyses of correlations between clinicopathological features and lung-cancer-specific survival for patients with pT2a (<3 cm, with pleural invasion) N0 M0 NSCLC that underwent sublobar resection or lobectomy.

Variables	Lung-Cancer-Specific Mortality					
	Univariate Analysis			Multivariate Analysis		
	Hazard Ratio	95% CI	*p* Value	Hazard Ratio	95% CI	*p* Value
Surgical method						
Sublobar resection						
Lobectomy	0.66	0.47–0.91	0.011	0.75	0.52–1.08	0.124
Smoking						
Never smoker						
Ever smoker	2.32	1.72–3.13	<0.001	1.43	0.96–2.12	0.077
Age						
≤75 yrs						
>75 yrs	2.74	1.99–3.77	<0.001	1.97	1.40–2.77	<0.001
Sex						
Male						
Female	0.51	0.38–0.69	<0.001	0.80	0.54–1.18	0.259
Laterality						
Right						
Left	0.95	0.70–1.29	0.726	0.82	0.60–1.13	0.231
Differentiation						
Well						
Moderate	1.66	0.84–3.29	0.147	1.31	0.66–2.60	0.446
Poor	3.47	1.73–6.94	<0.001	1.99	0.97–4.06	0.060
Tumor size						
0–1 cm						
1–2 cm	1.84	0.67–5.09	0.240	1.51	0.54–4.21	0.430
2–3 cm	2.76	1.02–7.48	0.046	2.19	0.80–6.01	0.129
Histology						
SCC						
Adenocarcinoma	0.32	0.20–0.51	<0.001	0.60	0.36–1.01	0.055
Others	0.88	0.47–1.64	0.687	0.91	0.48–1.74	0.785
Visceral pleural invasion						
PL1						
PL2	2.28	1.69–3.09	<0.001	2.04	1.50–2.77	<0.001
Lymphovascular invasion						
No						
Yes	8.15	4.54–14.63	<0.001	5.21	2.84–9.57	<0.001
Lymph node dissection number						
≥6						
<6	2.17	1.45–3.25	<0.001	1.51	0.97–2.35	0.067

CI, confidence interval; SCC, squamous cell carcinoma.

## Data Availability

Restrictions apply to the availability of these data. Data were obtained from Taiwan Cancer Registry Database and are available with the permission of the Taiwan Ministry of Health and Welfare.

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
