# Peer review of "Sublobar Resection Versus Lobectomy for Small (≤3 cm) NSCLC with Visceral Pleural Invasion: A Propensity-Score-Matched Survival Analysis from a Nationwide Cohort"

_cancers, 2025, doi:10.3390/cancers17121990_

Round 1

Reviewer 1 Report

Comments and Suggestions for Authors

This study investigated the comparative efficacy of sublobar resection (SLR) versus lobectomy in patients with non-small cell lung cancer (NSCLC) measuring ≤3 cm and presenting with visceral pleural invasion (VPI). The study was meticulously designed, featuring a large sample size and robust statistical analysis, thereby offering a valuable reference for clinical decision-making. Nevertheless, there are still certain methodological concerns and data limitations that warrant further discussion.

  1. The subgroups of wedge resection and segmentectomy, as well as the use of preoperative neoadjuvant therapy, are crucial for interpreting the study results.  To enhance the credibility of the article, it would be highly beneficial if the authors could provide additional subgroup analyses.
  2. After PSM, the number of lymph nodes removed in the lobectomy group was significantly higher than that in the SLR group (19.29 vs. 12.99, p<0.001). This difference may impact the accuracy of staging and the observed survival differences. Further analyses are recommended.
  3. The references cited are somewhat dated, with a relatively low proportion of sources from the past three years. It would be advisable for authors to incorporate more recent research materials to enhance the timeliness and relevance of the work.

Reviewer 2 Report

Comments and Suggestions for Authors

Lung cancer from the perspective of surgical curability is an area of ​​great interest in the medical world. It is known that in stages I and II, and in stages III a and b, these surgical approaches have a curative role.
I read with great interest the article sent from the perspective of surgical operations with the avoidance of radical methods.
I like how the article is structured and how the results and conclusions presented by the authors are in line with the proposed research objectives.
The methods used are clear and the statistical analysis is well chosen.

Comments on the Quality of English Language

The originality of the study is that it demonstrates similar results between radical surgical methods (lobectomy) compared to more limited surgical methods (sublobar resection).
The results presented encourage further studies.

Reviewer 3 Report

Comments and Suggestions for Authors

Dear authors, congratulations for the work presented.
A very interesting topic for both oncological and minimally invasive development. There are several limitations to the work, but well argued and above all not so decisive.
Good exposition and method. Certainly fundamental surgical data are missing, but this would go outside the focus of the work.
Excellent starting point for further studies.
